# Risk of introduction and establishment of alien vertebrate species in transboundary neighboring areas

Qing Zhang[1,2], Yanping Wang [1] ✉ & Xuan Liu [2,3] ✉

Cross-border neighboring areas could be particularly vulnerable to biological invasions due to short geographic distances and frequent interactions, although the invasion risk remains unevaluated worldwide. Here, based on global datasets of distributions of established alien vertebrates as well as vectors of introduction and establishment, we show that more than one-third of the world's transboundary neighboring areas are facing high invasion risk of alien vertebrates, especially in Europe, North America, South Asia, and Southeast Asia. The most important predictors of high introduction and establishment risk are bilateral trade, habitat disturbance and the richness of established alien vertebrates. Interestingly, we found that border fences may have limited effects in reducing the risk, as only 7.9% of border fences spatially overlap with hotspots of biological invasion even in the Eurasia areas (13.7% overlap) where physical border barriers are mainly located. We therefore recommend the implementation of immediate and proactive prevention and control measures to cope with cross-border invasions in response to continued globalization.

Invasive alien species (IAS) are a significant contributor to global environmental change, causing substantial declines in biodiversity, economic losses, and threats to public health[1-4]. To achieve the ambitious goal of conserving global biodiversity by 2050, the Post-2020 Global Biodiversity Framework under the Convention on Biological Diversity (CBD) has called for greater efforts by Parties to reduce the impacts of IAS[5]. Quantifying the risks of biological invasion in vulnerable areas has been recognized as one of the most effective strategies to prevent the increasing rate of IAS under intensified globalization[6,7]

Country borders are among such vulnerable areas with considerable academic and public concerns about environmental challenges in recent years[8-10]. The construction of transboundary infrastructure accelerated by regional economic development can promote trade and transportation, all of which can facilitate alien species invasions, making country borders especially sensitive to IAS[11]. In addition, cross-border countries often share corridors (e.g., rivers, roads, etc.) promoting alien species introduction. For example, weed invasion in Austria and Hungary is likely due to their well-connected transportation systems and frequent agricultural activities[12], while forest pests enter the eastern United States through heavily trafficked crossings along the Canada-U.S. border[13]. Neighboring areas are also susceptible to present similar environmental niches, which might facilitate the establishment of species in adjacent countries. For instance, analyses suggest that once an alien species has been established on one side, there is a high chance that it will invade neighboring countries without effective biosafety strategies[14]. Geopolitical forces such as border fences, contrasting national policies and regional conflicts can have both positive and negative relationships with alien species invasions, and thus may all complicate the efficiency of IAS

[1]Laboratory of Island Biogeography and Conservation Biology, College of Life Sciences, Nanjing Normal University, Nanjing 210023 Jiangsu, China. [2]Key Laboratory of Animal Ecology and Conservation Biology, Institute of Zoology, Chinese Academy of Sciences, 1 Beichen West Road, Chaoyang 100101 Beijing, China. [3]University of Chinese Academy of Sciences, 100049 Beijing, China. ✉e-mail: wangyanping@njnu.edu.cn; liuxuan@ioz.ac.cn

coordination strategies. For example, conflicts can promote the transportation of alien species through military material[15], and contrasting national policies can make it difficult to coordinate IAS management strategies between countries[14]. All these factors increase the uncertainty of cross-border invasions. Therefore, identifying biological invasion hotspots in transboundary neighboring areas is crucial for early prevention and strict surveillance, ensuring the sustainable development of cross-border regions[16,17].

To accurately assess the risks of alien species invasions in transboundary neighboring areas, it is vital to quantify the primary drivers of alien species introduction and establishment[18]. Anthropogenic factors, including bilateral trade, traffic transportation, and human movement, play a crucial role in facilitating the introduction of IAS. For instance, trade is a primary pathway for IAS introduction, transporting stowaways or pollutants in goods and packaging materials. These imported species may unintentionally or intentionally escape or be released into the wild[19,20]. Geopolitical boundaries are a particular concern, as the wildlife trade is rampant in these areas[21]. The rapid development of regional infrastructure and transboundary transportation routes may also aid IAS dispersal across political borders via vehicles[22]. Additionally, landscape geomorphology, such as cross-border river systems, is increasingly recognized as important for the dispersal of alien propagules from neighboring countries[23]. For example, there are a total of 153 countries sharing transboundary rivers, providing IAS dispersal passageways, particularly for aquatic species[24]. However, geographic barriers may also exist in areas separated by mountains, limiting IAS dispersal[25]. After introduction, propagule pressure is crucial determinant of alien species establishment[26,27]. More introduction events and individuals in each introduction event can provide higher propagule pressure and increase the likelihood of establishment[28]. Meanwhile, transboundary neighboring areas are also facing high pressures of habitat disturbances due to the increasing infrastructure construction in the border areas[8]. The habitat disturbance hypothesis predicts that alien species may be more likely to establish populations and rapidly disperse in disturbed habitats by creating more vacant niches[29,30] after removing native predators and competitors[31]. Furthermore, more existing invaders can also facilitate subsequent invasions through the process of invasion meltdown[32,33]. Species environmental suitability is also regarded as important for the establishment of alien species, although it is dependent on species-specific analyses, and the complete listing knowledge of potential alien species and their ecological requirements are usually lacking at large spatial scales[34]. However, a comprehensive evaluation of the relative contribution of these potential factors in predicting the biological invasion risk in transboundary neighboring areas worldwide is still lacking.

Here, we provided a global-scale study to identify the invasion hotspots of alien vertebrate species in terrestrial transboundary neighboring areas. This was achieved by first conducting risk analyses for the introduction and establishment of alien vertebrate species that were then integrated to obtain the overall invasion risk (for details, see Methods, refer to Supplementary Fig. 1). Furthermore, we explored the relative importance of different environmental and anthropogenic variables, such as river network density (RND), traffic network density (TND), bilateral trade volume (BTV), land use change frequency (LUCF), and richness of established alien vertebrates (REA), in predicting the potential invasion risk. Finally, we compared the spatial relationship between regions with very high overall risk and bilateral cooperation abilities as well as physical border barrier distributions, which may further complicate neighboring transboundary invasion risk. Our findings can indicate those neighboring regions where transboundary collaborations should be a priority aimed at fortifying early and effective intervention against alien vertebrate invasions through coordinated management.

## Results

### Global transboundary invasion risk

We evaluated the introduction, establishment and overall risk in transboundary neighboring areas by analyzing 334 bilateral borders, which intersected with a total of 5,088 0.5° grids. After ranking the grids based on different introduction and establishment vectors, we assigned five levels from very low (VL) risk to very high (VH) risk for each grid. Then we defined the introduction and establishment hotspots in transboundary neighboring areas as those top 20% grids with the highest level of risk (i.e., the VH level) posed by any one introduction or establishment vector (Fig. 1). Our results showed that the proportions of hotspots for introduction and establishment were 43.01% and 31.89%, respectively. We overlapped the introduction and establishment risks for each grid (Supplementary Fig. 1), which revealed that more than one-third (36.3%, 1,845 grids) of global borders were identified as overall invasion hotspots with high introduction and establish risk simultaneously, with a concentration in North America, Europe, South Asia and Southeast Asia (Fig. 2).

Approximately 74.6% of borders (249/334) contained at least one grid identified as an overall invasion hotspot. Among them, the Canada-U.S. border held the highest number of invasion hotspots in terms of introduction (Fig. 1), establishment (Fig. 1), and overall risks (Fig. 2). Additionally, we found that the top five borders with the highest number of introduction hotspots are Canada-U.S., Kazakhstan-Russia, Norway-Sweden, Bangladesh-India, and Argentina-Chile (Fig. 1a). The establishment hotspots were primarily concentrated in the Canada-U.S., Mexico-U.S., China-India, India-Nepal, and India-Pakistan borders (Fig. 1b). We observed that Canada-U.S., Kazakhstan-Russia, Mexico-U.S., India-Nepal, and Bangladesh-India ranked among the top five bilateral borders with a higher number of overall invasion hotspots than other borders (Fig. 2).

To test whether the grid size and percentiles defining invasion hotspots may influence the results, we conducted sensitivity analysis using different grid-cell sizes (i.e., 0.5°, 1°) and different percentiles defining invasion hotspots (i.e., >90%, >80%, >70%), which obtained consistent spatial patterns, indicating that our results were robust to data uncertainties (Supplementary Fig. 2).

### The relative contribution of different variables predicting invasion hotspots in cross-border regions

We further identified the relative importance of different introduction factors (i.e., RND, TND, BTV) and establishment factors (i.e., LUCF and REA) in predicting the overall invasion hotspots (Supplementary Data 1). Our results showed that BTV, LUCF, and REA were identified as the top variables (Kruskal−Wallis test, $H = 110.58$, $P < 0.01$, Supplementary Fig. 3) contributed to overall invasion hotspots (Fig. 3, Supplementary Table 1). Approximately 80.5% of overall invasion hotspots were generated by at least three factors (Supplementary Table 1, Supplementary Data 1), and further null-model tests showed that overall invasion hotspots were more concentrated on Europe and South Asia borders than expected by chance (Mann–Whitney U test, $Z = −1.73$, $P < 0.05$, Fig. 3, Supplementary Table 1, Supplementary Figures 4–6).

We identified that the primary factors that are associated with invasion hotspots vary greatly across border regions (Fig. 4). Among the top five introduction hotspots, we observed that the risk at the Canada-U.S. and Norway-Sweden borders largely resulted from very high levels of bilateral trade volume (BTV). Comparatively, the primary factor driving introduction risk at the Kazakhstan-Russia border was traffic network density (TND) (Fig. 4a). Among the top five establishment hotspots, we found that the very high risk of establishment at the Canada-U.S. and Mexico-U.S. borders was largely attributable to the richness of established alien vertebrates (REA), whereas the establishment risk at the India-Pakistan border was mainly shaped by land use change frequency (LUCF) (Fig. 4b). Among the top five overall

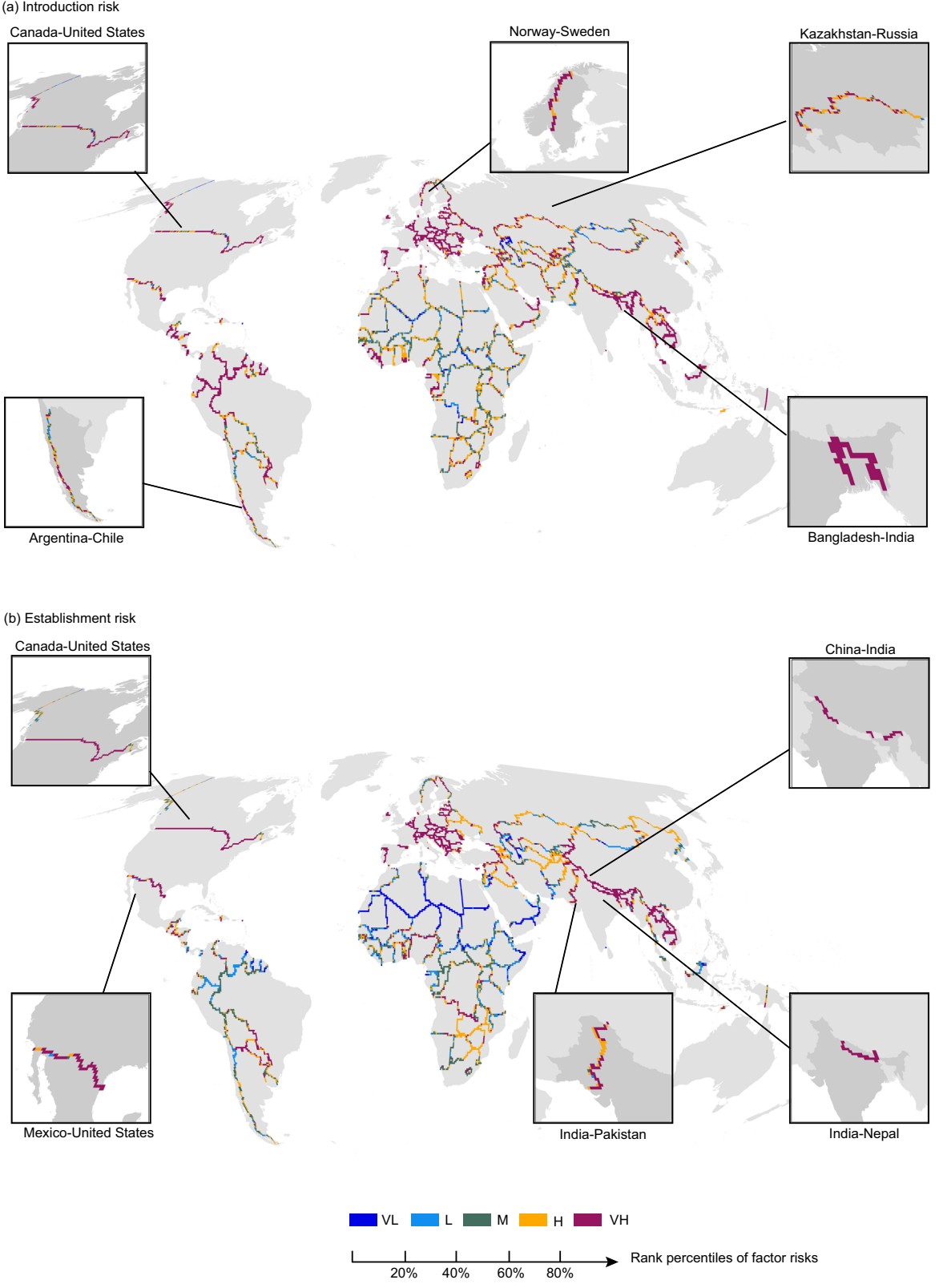

**Fig. 1 | Introduction and establishment risks in transboundary neighboring area worldwide.** This figure presents an assessment of **a** introduction and **b** establishment risks in cross-border areas by integrating river network density (RND), traffic network density (TND), bilateral trade volume (BTV), the richness of established alien vertebrates (REA) and the degree of habitat disturbance-land use change frequency (LUCF). To assign the introduction and establishment risk level, the higher level of introduction or establishment factors in each grid were considered using the nonadditive method. The grey shadowing shows the global map except Antarctica. The panels also indicate the top five pairs of bilateral countries colored with dark grey shadowing with the largest number of introduction and establishment hotspots. VL very low, L low, M medium, H high, and VH very high.

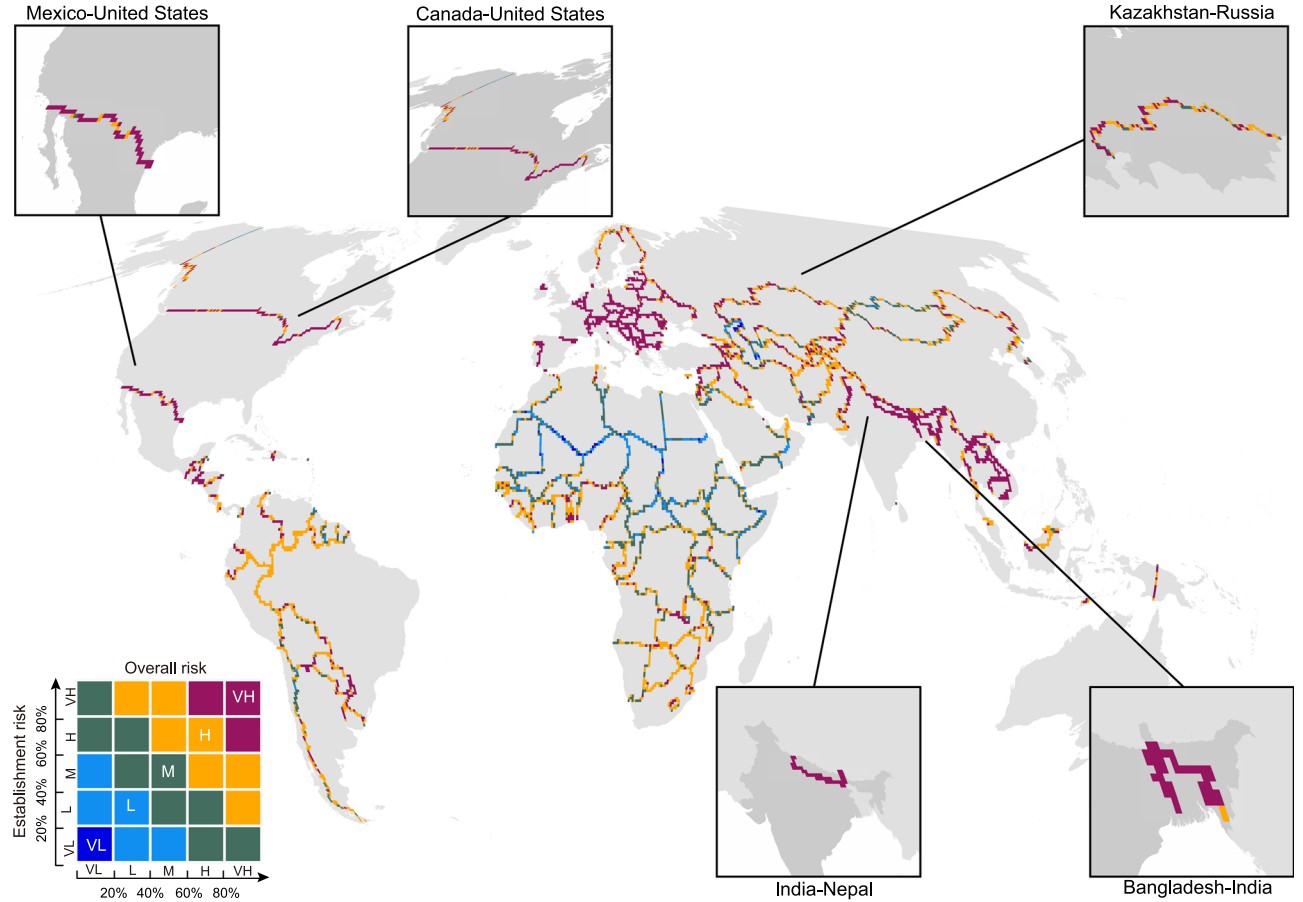

**Fig. 2 | Overall invasion risk in transboundary neighboring area worldwide.** Introduction and establishment risks (Fig. 1) are combined to calculate the overall invasion risk in neighboring cross-border areas. The panel is presented using the same color scheme as in Fig. 1. The panel also indicates the top five bilateral countries with the largest number of overall invasion hotspots. VL very low, L low, M medium, H high, and VH very high.

invasion hotspots, we found that bilateral trade volume (BTV) was the key factor contributing to the very high overall risk at the Mexico-U.S. borders, while river network density (RND) was the major factor driving the very high overall risk at the India-Nepal and Bangladesh-India borders (Fig. 4c).

We found that there have been many established alien vertebrates recorded in transboundary neighboring areas across the globe (Supplementary Data 2), which contributed to 72.6% of the overall risk (Supplementary Table 1). There were significant taxonomic and geographical variations in the number of established alien vertebrates. Alien fish, bird, and mammalian species were found to be more pervasive than alien amphibian and reptile species (Kruskal–Wallis test, $H = 622.68$, $P < 0.01$; Supplementary Fig. 7). Overall, the transboundary neighboring areas with a high number of established alien vertebrates were largely concentrated in Europe ($H = 147.25$, $P < 0.01$, Supplementary Figs. 8 and 9). To validate whether the result may be sensitive to sampling bias of IAS distribution data, we also calculated the richness of established alien vertebrates after accounting for the sampling effort (for details, see Methods) and did not find a significant difference in the number of overall invasion hotspots for neighborhood borders (two-tailed Wilcoxon rank sum test, $Z = -43.88$, $P > 0.05$, Supplementary Fig. 10), indicating that our results were robust to potential data sampling biases.

Specifically, established alien fish species were mainly distributed in North America, such as the Great Lakes region, southeastern Europe, Paraguay, and some South and Southeast Asian countries, such as India-Nepal and Laos-Thailand, and China with Central Asian countries (Fig. 5a, Supplementary Data 2). Established

alien birds were prevalent in the Canada-U.S. and Mexico-U.S. borders in North America, the Lesotho-South Africa border in southern Africa, Argentina-Chile, and Brazil-Colombia borders in South America and Europe (Fig. 5d, Supplementary Data 2). Established alien mammals were mostly located in the Eurasian continent, including the borders along Russia, Germany, France, Poland, Finland, Moldovan, Ukraine and Belarus, and China-Kazakhstan and Indonesia-Papua New Guinea, as well as Argentina-Chile in South America and Canada-U.S. in North America (Fig. 5e, Supplementary Data 2). Established alien amphibians and reptiles widely distributed at the Mexico-U.S., Europe, and southern Africa borders (Figs. 5b, c, Supplementary Data 2).

## Uncertainty of cross-border invasion risks

We investigated the relationship between invasion risk in neighboring cross-border regions and two potential confounding factors namely cross-border cooperation capacity and physical barriers. Specifically, cooperation, governance, and human pressure were combined to quantify cross-border cooperation capacity[9]. Information on physical barriers across the world was gathered from different sources (for details, see Methods), which are mostly distributed in Eurasia[10] (Supplementary Fig. 11). We found that there was no significant relationship between the number of overall invasion hotspots and cross-border cooperation capacity (linear regression, $R^2 = 0.0046$, Fig. 6a, $P > 0.05$) or physical barriers (Mann–Whitney U test, $Z = -0.90$, Fig. 6b, $P > 0.05$). For example, some areas with low bilateral cooperation capacities, such as the India-Nepal and Bangladesh-India borders, were quantified as very high overall risk areas (Supplementary Fig. 12a). Furthermore,

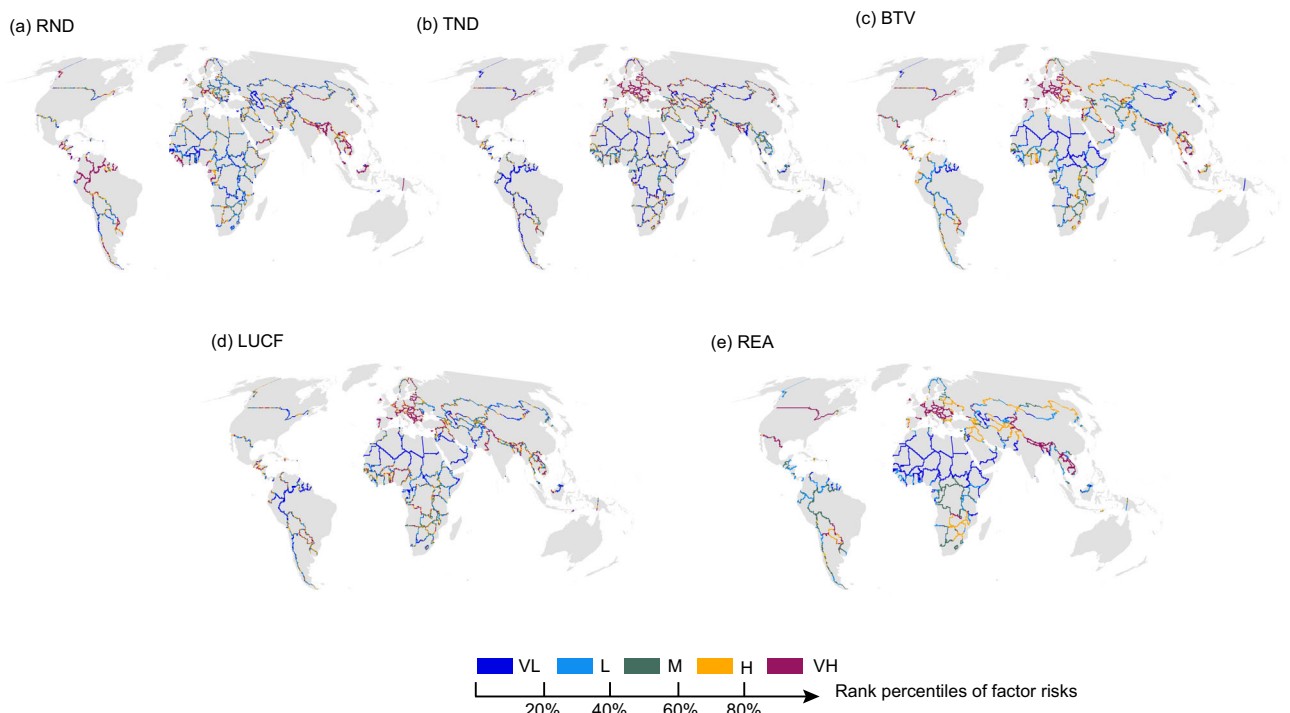

**Fig. 3 | Primary factors predicting introduction, establishment, and overall invasion risks of cross-border areas worldwide.** Panels **a**–**c** illustrate the three factors (RND, TND, BTV) that contribute the most to introduction risk, while panels **d** and **e** show the two factors (LUCF, REA) that contribute the most to establishment risk. RND: river network density, TND: traffic network density, BTV: bilateral trade volume. LUCF: land use change frequency, REA: richness of established alien vertebrates. All panels are presented using the same color scheme as in Fig. 1. VL very low, L low, M medium, H high, and VH very high.

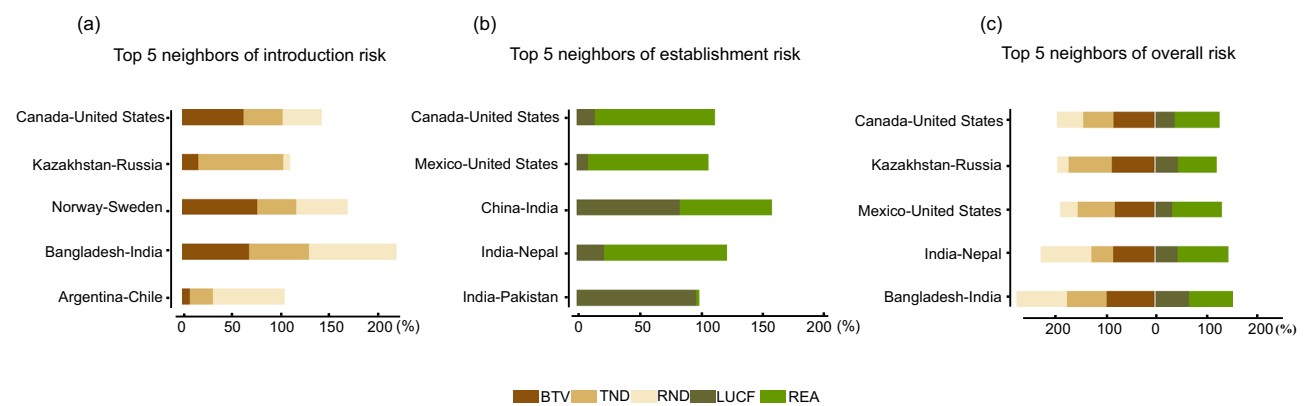

**Fig. 4 | The ratio of very high-risk grids (VH level) for different predictor variables along the top five neighbors of (a) introduction risk, (b) establishment risk, and (c) overall risk.** These predictors include bilateral trade volume (BTV), traffic network density (TND), river network density (RND), land use change frequency (LUCF), and the richness of established alien vertebrates (REA). Note that as some grids may have different VH factors, the summation of VH grid ratios across different factors may exceed 100%.

many cross-border regions, such as Laos-Vietnam and Portugal-Spain, with very high overall risk (approximately 77.1% of grids with very high risk, as shown in Figs. 2 and 6, Supplementary Fig. 12b), were not covered by physical barriers.

## Discussion

Our present study focusing on the risk of alien vertebrate invasions in global transboundary neighboring areas contributes to the early prevention of alien species invasion risks compared with previous studies that mostly focused on risk analysis at the whole country level[14,34,35]. We found that previous invasion hotspots across Europe[14,36], southern or southeast Asia[17], the North American Great Lakes[37], and the Amazon,

Ganges, and Mekong basins[38] also face high invasion risks in cross-border areas. These areas were distinguished by their frequent bilateral economic and transportation activities, connected landscapes, and high richness of established alien species.

Our study indicated that there were geographical variations in factors explaining introduction, establishment and overall risks across different transboundary neighboring areas. In Europe, we observed a high risk from established alien vertebrates, bilateral trade, transportation, and habitat disturbance. The high cross-border invasion risks in Europe were likely due to the ever-expanding form of integrated regimes[39,40] and the removal of physical barriers[12]. One potential issue in quantifying the spatial distributions of introduction and

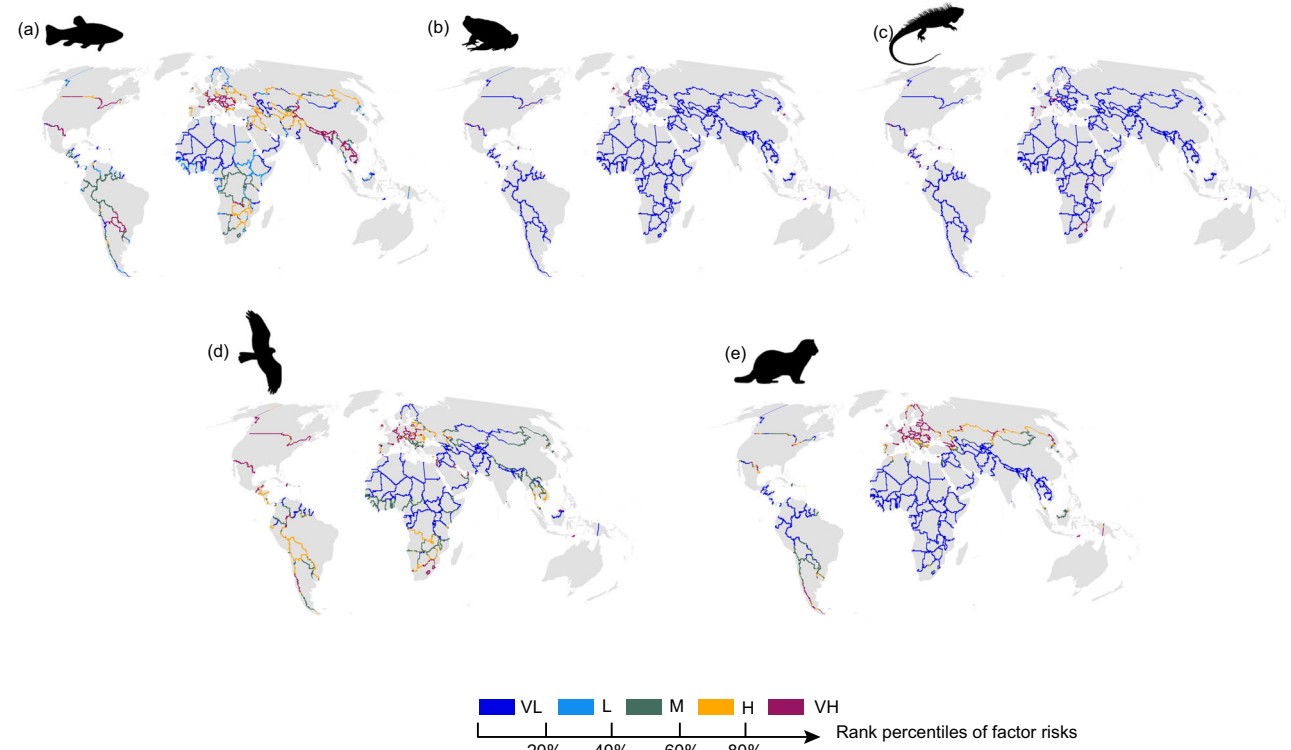

**Fig. 5 | Geographical variations in the number of established alien vertebrates across global borders. a** Fish, **b** Amphibian, **c** Reptile, **d** Bird, and **e** Mammal. All panels are presented using the same color scheme as Fig. 1 for consistency. Species silhouettes are sourced from PhyloPic (http://www.phylopic.org/). VL very low, L low, M medium, H high, and VH very high.

establishment risk at country borders is that it might be related to the length of border lines, area and number of neighboring countries among continents. We therefore conducted supplementary analyses and did not detect a significant relationship between the number of overall invasion hotspots and the number of countries (Spearman correlation coefficient $r = 0.5$, $P > 0.05$), the length of borders ($r = 0.5$, $P > 0.05$) or the country size ($r = 0.3$, $P > 0.05$).

With high traffic pressures, there should be a higher level of screening for suspected species introduced through roads, railways, and waterways[38]. While there had been national-level legislation and relevant international conventions, such as CITES and trade restrictions imposed by bilateral national agreements to prevent alien species introductions, cross-border regions were also hotspots for illegal wildlife trade[41], which was a pathway facilitating the introduction of alien species[42]. For example, the Mexico-U.S., China-Myanmar, Bangladesh-India, and some Asian borders experience high pressure to prevent illegal wildlife trade[17,43] which were also identified as transboundary invasion hotspots in our study. We need to pay close attention to areas undergoing rapid anthropogenic disturbances across the African, American and Asia-Pacific tropics, such as Brazil-Paraguay, Colombia-Venezuela, and Indonesian Borneo[44,45], which were all identified as transboundary invasion hotspots (Figs. 1–3; Supplementary Table 1, Supplementary Data 1). For regions with unobstructed landscape connectivity, it was necessary to focus on ballast water management and identify locations where riverbank stability and water quality were deteriorating, which may facilitate invasions[46]. Areas with large numbers of established alien species could generate high prevention pressures of existing invaders and might facilitate the establishment of new alien species according to the invasion-meltdown hypothesis[33]. Therefore, for neighboring countries with a high richness of established alien vertebrates in Europe and North America, South and Southeast Asia, parts of the border in Central Asia, Western Asia, and Southern Africa, timely IAS information

sharing and formulated prevention and control strategies for existing and future potential IAS, as well as strengthened laws and regulations to restrict alien species bilateral import and export[47], are needed. We also found taxonomic variations in the number of established alien vertebrates among global transboundary neighboring areas (Fig. 5). Fishes ranked the top taxa with the largest number of established alien vertebrates, largely due to aquatic species' faster dispersal capacity than terrestrial species[48]. Although the distributions of established alien vertebrates have not been surveyed systematically across the globe, we obtained similar results with the main analysis after accounting for the sampling bias issue, demonstrating the robustness of our results to data uncertainties (Supplementary Fig. 10). Moreover, the role of different introduction pathways may be dependent on the exact taxa. For instance, river networks will be particularly important for taxa such as fishes, amphibians and reptiles with life stages in the water and some birds that use water bodies to promote dispersal[49]. Mountainous topographic heterogeneity (MTH) might be more likely to act as a geographical barrier for taxa with relatively low natural dispersal abilities[25]. We therefore conducted supplementary analyses using the exact important pathways for certain taxa and obtained similar risk patterns (Supplementary Fig. 13).

One effective approach for preventing IAS was bilateral cooperation between neighboring countries. However, our study found that cross-borders with high cooperation did not match the invasion hotspot locations. Indeed, only a few bilateral countries have high capabilities to manage invasion risks[50], and there are unbalanced capacities and differentiated policies among neighboring countries in resisting invasions[51]. In addition, we found that physical barriers might not be able to limit cross-border invasions. We obtained consistent results when we focused our analyses in Eurasia where physical border barriers were mainly located (Supplementary Fig. 14, Mann–Whitney U test, $Z = -4.74$, $P > 0.05$) and when we conducted analyses using only alien reptiles and mammals, the dispersal of which might be

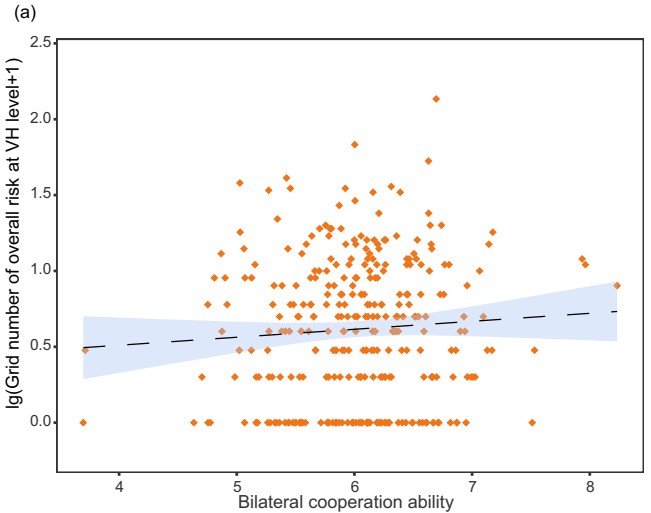
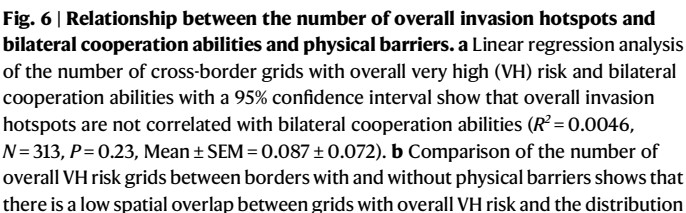
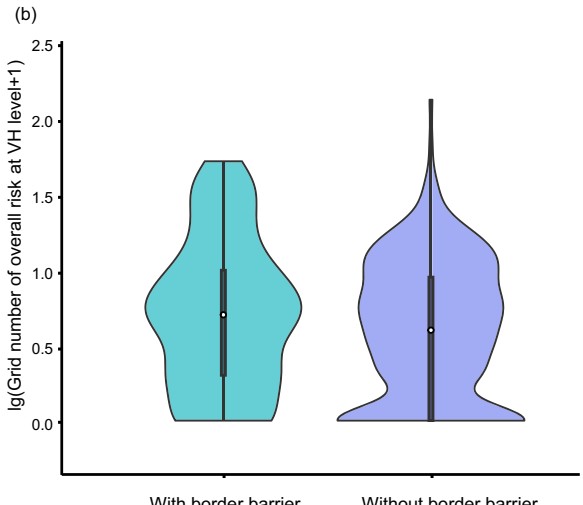

**Fig. 6 | Relationship between the number of overall invasion hotspots and bilateral cooperation abilities and physical barriers. a** Linear regression analysis of the number of cross-border grids with overall very high (VH) risk and bilateral cooperation abilities with a 95% confidence interval show that overall invasion hotspots are not correlated with bilateral cooperation abilities ($R^2 = 0.0046$, $N = 313$, $P = 0.23$, Mean ± SEM = 0.087 ± 0.072). **b** Comparison of the number of overall VH risk grids between borders with and without physical barriers shows that there is a low spatial overlap between grids with overall VH risk and the distribution of physical barriers (two-sided Mann–Whitney U test, $Z = -0.90$, $P > 0.05$, $n_{without\ border\ barrier} = 265$, $n_{with\ border\ barrier} = 48$). The medians of the box plots showing VH risk grids between borders with and without physical barriers are 0.70 and 0.60, respectively. The IQR of the box plot for borders with physical barriers is 0.70, while that for borders without physical barriers is 0.95. In the boxplots, the white dot is the median, the black box type ranges from the lower quartile to the upper quartile, and the thin black line indicates the whisker. Global border physical barriers are presented in Supplementary Fig. 11.

particularly influenced by physical barriers (Supplementary Fig. 15, $Z = -0.80$, $P > 0.05$). Border fences had mostly been designed for political and social security reasons rather than biodiversity conservation[52]. Thus, the role of physical borders in preventing IAS warrants further investigation, especially as the number of border fences increases with regional political tensions[53].

We recognize that there were still some limitations of our present analyses. One concern was that alien species might not always spread through national boundaries but could also undergo long-distance dispersal through transportation by airplanes, ships, and even wars[15]. Our present study focused on neighboring border areas, which are theoretically sensitive to biological invasions due to their shared connectivity corridors, similar climate, frequent trade and economic exchanges[8–10], but have also been a neglected aspect of invasion risk analyses in literatures. Second, it is known that habitat suitability is an important determinant of alien species establishment. As habitat suitability needs species-specific analyses by incorporating all potential alien species in the transboundary areas and their ecological requirements, which is not yet available at the global scale[34], we did not conduct such analyses but suggested that it might be an avenue for future studies. In addition, our present analyses focused on established alien vertebrates, but it is important to note that there were also increasing introductions of other alien species[54,55], highlighting the necessity of controlling the risks from more future potential invaders in different taxa. This was particularly true under the growing global transportation of goods and trade flow, which could introduce new alien species across taxa[34], and rapid development pressures could further lead to habitat disturbance and promote invasions[35]. Fortunately, there were emerging programs to address this issue, such as the Caribbean Invasive Species Project and the Working for Water (WfW) program in South Africa[50], which were devising relevant policies to prevent bilateral invasions.

In conclusion, our study suggested that approximately one-third of cross-border neighboring areas might be considered invasion hotspots worldwide. Factors predicting invasion risks varied across global borders, and thus different control measures adjusting to local conditions were needed based on the exact risk factors. The low spatial

correlation of physical barriers or cooperation capabilities with invasion hotspots demonstrated the need for proactive planning and closer cross-border bilateral cooperation to effectively manage invasion risks in transboundary neighboring areas. Our present study takes an essential step toward quantifying invasion hotspots along the cross-borders, which might assist in the development of united and timely management and control efforts to prevent biological invasions in transboundary neighboring areas during the era of globalization.

## Methods

We used a quantitative framework that combined the relative likelihood of introduction and establishment of alien vertebrate species to evaluate the invasion risks in transboundary neighboring areas. The framework was based upon several previous studies[18,26,29,34] and its graphical representation is shown in Supplementary Fig. 1.

### Global transboundary maps

We used data from the Global Database of Administrative Areas v.3.4 (GADM, 2020) and ministry of natural resources standard mapping service (http://bzdt.ch.mnr.gov.cn/; No. GS (2016) 1666) to determine global administrative geographic ranges and their corresponding terrestrial border lines. Our analyses excluded Antarctic areas, Australia and New Zealand that do not share terrestrial neighboring borders. All introduction and establishment predictor variables were extracted, and analyses were conducted at a spatial resolution of 0.5° grids intersecting the border lines of neighboring countries under the Mollweide projection using ESRI ArcGIS Pro v.2.5.2. This spatial resolution was widely accepted as appropriate for global-scale studies because it balanced analysis precision and computational efficiency. Additionally, it was considered reasonable for making invasion biosecurity decisions at large spatial scales[34].

### Introduction risk factors

The introduction risk across borders depended on several factors, including trade volume, frequency of transport, human movement, and the nature of the cross-border landscape[19,20,26]. In terms of trade volume, the number of alien species introduced intentionally (for

aquaculture, horticulture, scientific research, biological control, etc.) and unintentionally (as pests and pathogens on food, packaging materials, or livestock, or as stowaways and contaminants of crop seed) was directly proportional to the volume of trade[56,57]. The transportation frequencies and human movement associated with trade were also significant determinants of alien species introduction, as trade commodities typically circulate through high traffic and densely populated areas[57,58].

To investigate the introduction risk through trade, we examined bilateral trade volumes using mean annual U.S. dollar values of imported and exported goods between each pair of countries from 2011 to 2020, obtained from the United Nations Commodity Trade Statistics Database (https://comtrade.un.org/, accessed in September 2021). Since trade data were only available at the country level and the introduction risk depended on the final traded commodity destinations, which were associated with the distribution of local human populations, we used an introduction epicenter method[34] to estimate the introduction risk by bilateral trade for each cross-border grid. This involved first calculating the bilateral trade volume per capita by dividing the total quantity of trade by the human population size for the transboundary countries. We then calculated the trade induced introduction risk by multiplying the per capita trade by the human population density of each cross-border grid. We collected human population data from the United Nations database (https://population.un.org/wpp/Download/Standard/Population) and the NASA database (Gridded Population of the World (GPW), v4; https://earthdata.nasa.gov, 2020) at a resolution of 2.5 arcmin and resampled it to a 0.5° grid.

As a crucial mode of alien species introduction in transboundary neighboring areas, short-distance smuggling of vehicles[22] and road construction[59] were frequently observed. To assess the introduction risk via the transportation network, we calculated the length of roads and railways within the cross-border grid[60]. The data for roads and railways were acquired from the Natural Earth database (https://www.naturalearthdata.com, accessed in September 2021). Moreover, although air flights were usually used as predictors related to IAS long-distance dispersal[61], we also collected the number of airline routes (ALR) from openflights.org (openflights.org, 2023)[14] for the airports closest to each grid of the transboundary neighboring areas and obtained similar patterns of the introduction risk with our main analysis (Supplementary Fig. 16). Furthermore, border areas are usually set along geographic barriers such as rivers, lakes, mountain ranges and high tableaus. Biogeographic landscapes such as river systems constitute vital corridors for IAS movement across administrative divisions[23]. The evaluation of introduction risk emanating from river distribution data obtained from the Mapping the world's free-flowing rivers database[62]. Considering that river networks may be particularly important for fishes and herpetofauna with life stages in aquatic environments and bird migration, we also conducted supplementary analyses specific to fishes, herpetofauna and birds (Supplementary Fig. 13a–d). Moreover, to quantify the effect of mountainous topographic heterogeneity (MTH), which may act as a barrier to IAS dispersal, we conducted supplementary analyses by including MTH for most taxa except birds with relatively high dispersal abilities (Supplementary Fig. 13a, b, c, e). We quantified the MTH by averaging the maximum range in elevation of all 30 arc-second grid cells within each 0.5° grid using data from WorldClim following previous studies[25,63], and the normalized values were used for further analyses.

## Establishment risk factors

The establishment of alien species was primarily determined by propagule pressure, invasion meltdown, and the level of habitat disturbance, as evident in previous studies[28]. Propagule pressure is a crucial factor that affects the establishment of alien species, but the availability of precise data on the number of introductions and individuals per event is often challenging. Therefore, past studies have typically used proxy variables to evaluate propagule pressure[28], such as the factors mentioned in the introduction risk analyses above. Furthermore, according to the invasion-meltdown theory[33], the presence of existing invasive species was a strong indicator that might significantly promote the establishment of subsequent alien species. To estimate the richness of established alien vertebrates in transboundary neighboring areas, we utilized cross-border grids to intersect the distribution ranges of each established alien vertebrate (361 fishes, 42 amphibians, 64 reptiles, 87 birds and 72 mammals). To achieve this goal, we conducted data collection from published databases and literature using a total of 12 major languages including English, French, Danish, Estonian, Finnish, German, Norwegian, Portuguese, Russian, Spanish, Swedish, and Mandarin Chinese, to account for the potential effect of sampling bias of established alien species distributions among continents[64] (Supplementary Data 3 and 5). Specifically, to assess the distribution of established alien birds and mammals, we used polygon data from two prominent global databases, including the Global Avian Invasions Atlas (GAVIA) and the global Distribution of Alien Mammals database (DAMA), which have incorporated the multilingual literature[65,66]. The information on the list of established alien amphibians and reptiles was mainly obtained from Kraus and Capinha[67,68], and we then collected the occurrence data of different established alien amphibians and reptiles from intensive literature reviews using different languages above (Supplementary Data 3). We collected the distribution of established alien fishes at the drainage basin level from a public database[69,70] and a literature review using the different lanuages described above (Supplementary Data 3). There is ample evidence indicating that habitat disturbances may increase the likelihood of alien species invasions[30,71]. This phenomenon occurs as habitat disturbances create empty spaces and favorable conditions for species establishment and facilitate dispersal corridors in transboundary neighboring areas. To quantify the degree of habitat disturbance in cross-border grids, we analyzed land use change frequency (LUCF) data from 1960 to 2019 provided by Winkler[72]. Using the ArcGIS zonal statistics function, we calculated the mean value of land use change for each cross-border grid.

## Quantification of introduction, establishment, and overall invasion risks

We applied the max-min normalized method to sort each introduction and establishment factor, resulting in a ranking of the risks of all grid cells across global cross-borders. These grid cells were binned into percentiles, where 100-80% were categorized as very high (VH), 80-60% as high (H), 60-40% as medium (M), 40-20% as low (L), and 20-0% as very low (VL), as per the refs. 34,35. To further determine high introduction or establishment risks, we adopted a nonadditive approach to assess threat factors[34]. This was done by assigning the highest level of risk faced by any one single factor, as illustrated in Supplementary Fig. 4. For example, if a grid cell had high (H) risk because of its bilateral trade volume (BTV) and medium (M) risk because of the traffic network density (TND) and river network density (RND), then it would be rated as high (H) introduction risk. Finally, the introduction risk and establishment risk were overlapped to obtain the overall risk following the scheme in Supplementary Fig. 1. For example, we obtained the medium (M) overall risk when high (H) introduction risk and low (L) establishment risk level overlapped, and we identified overall invasion hotspots (i.e., grids with the highest overall risk) when both introduction and establishment risks were at their highest levels. We used the Kruskal–Wallis test to explore the relative importance of different introduction and establishment factors in predicting overall invasion hotspots (1845 grids) along various borders by comparing the number of grids with risk in H and VH levels of different factors (Supplementary Data 1, Supplementary Fig. 3). We also identified the relative importance of the richness of established alien species (REA) among different taxa in contributing to the overall invasion hotspots by comparing the relative proportion of REA across taxa along each border

using the Kruskal–Wallis test (Supplementary Data 4, Supplementary Fig. 7). To test whether there are spatial overlaps among those important factors contributing to overall invasion hotspots, we applied a null model approach by generating 1000 random distributions of high risk of introduction and establishment grids to test whether the spatial overlap between observed introduction and establishment risks was more concentrated in certain hotspot regions than that would be expected by chance based on the Mann–Whitney U test.

Finally, we also carried out sensitivity analyses to test the reliability of our study based on different grid-cell sizes, such as 0.5° and 1°, and different factor percentiles defining the highest introduction or establishment risks (i.e., >90%, >80%, and >70%). We also tested the potential influence of sampling bias on our results by quantifying the richness of established alien species using a sampling-effort-corrected approach[73,74]. To achieve this, we used the native vertebrate inventory as a proxy of sampling efforts for the richness of recorded established alien vertebrates, assuming a positive correlation between the sampling efforts of exotic and native species[73]. We obtained the mean normalized taxon species richness for each 1° grid as cross-taxon REA. Then, we fitted a linear model of lg (cross-taxon REA + 1) as a function of lg (sampling effort+1). Both variables were scaled to their standard deviations. We extracted the residuals from the model as the sampling-effort-corrected REA for further analyses. We conducted the two-tailed Wilcoxon rank sum test to compare the number of overall invasion hotspots for each of paired country borders before and after incorporating the sampling bias.

### Uncertainty of neighboring cross-border invasion risks

To investigate the potential political uncertainties of transboundary invasion risks resulting from active cooperation relations and physical barriers, we conducted linear regression analysis with a 95% confidence interval (CI) to examine the relationship between the number of overall invasion hotspots and bilateral cooperation abilities and compared the number of overall invasion hotspots between areas with and without physical barriers using the Mann–Whitney U test. To quantify bilateral cooperation abilities, we used a parameter similar to a "Feasibility index," known as "Bilateral cooperation ability"[9], which took into account information on cooperation, governance, and human pressure. The feasibility index was quantified as between 0 and 10, with larger values indicating a higher bilateral cooperation ability, and was averaged for each country pairing. Physical barrier distribution data were sourced from previous global reports with either fully fenced or under construction status[10,53] (Supplementary Fig. 11). As there are geographical variations in spatial distributions of physical border barriers that are mainly located in Eurasia and the effect of physical barriers may be particularly important for alien reptiles and mammals[8,10], we also conducted supplementary analyses specific to the Eurasia areas (Supplementary Fig. 14) and the alien reptile and mammal species (Supplementary Fig. 15), respectively. To enhance normality, the number of overall invasion hotspots per border was log transformed before statistical analysis.

### Reporting summary

Further information on research design is available in the Nature Portfolio Reporting Summary linked to this article.

## Data availability

The processed figure data, database and literature used to collect the occurrence data of established alien vertebrate species, and keywords for multilingual literature search to collect occurrence data of alien amphibians, reptiles and fish species are available at Supplementary Data. The map data[75] generated in this study using ESRI ArcGIS Pro v.2.5.2. that are available in the Figshare database (https://doi.org/10.6084/m9.figshare.24764388). Source data are provided with this paper.

## Code availability

Data analysis and plotting were processed with the "rstatix", "ggpubr", "ggplot2", "tidyverse", "ggthemes", "viridis", "hrbrthemes", "readr", "ggsignif", "coin", "PMCMR", "dplyr", "rsq", "raster", "spatialEco", and "sf", "PMCMRplus" packages in R v.4.1.1[76].

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

## Acknowledgements

We thank Yuanbao Du, Shimin Gu, Weishan Tu, Shengnan Chen, Yanhua Hong, Lixia Han, Yanxia Li, Jiajie Yu, Sadia Ashraf and Zhiqiang Lin for assistance with data collection. We also thank Akbayan Yerlankyzy, Santiago Montero-Mendieta and Wenjie Li for assistance with keywords in literature search using different languages. This work was supported by the Third Xinjiang Scientific Expedition Program (Grant No. 2022xjkk0800 to X.L.), the National Natural Sciences Foundation of China (Grant No. 32171657 to X.L., Grant No. 32271734 to Y.W.), the grants of high quality economic and social development in southern Xinjiang (NFS2101 to X.L.), and the grants from Youth Innovation Promotion Association of Chinese Academy of Sciences (Y201920) to X.L.

## Author contributions

X.L. is the lead contact for this manuscript. X.L. conceived the study; XL and YW supervised the project; X.L. and Q.Z. designed the study; Q.Z., X.L., and Y.W. collected and analyzed the data; Q.Z., X.L., and Y.W. wrote the manuscript.

## Competing interests

The authors declare no competing interests.
