## [Peer Review File · Nature Communications]

Risk of introduction and establishment of alien vertebrate species in transboundary neighboring areasREVIEWER COMMENTS

Reviewer #1 (Remarks to the Author):

Cross-border invasions have been a neglected aspect of such phenomena because each country takes care of its own problems as if it were completely isolated from neighbors or from globalized commerce. This is reflected in the lack of coordinated international efforts to monitor or control alien species that expand their ranges across borders. This paper thus addresses a novel, synthetic, timely, and welcome addition to the transboundary invasion problem. I am impressed at the worldwide and multi-species information amassed and analyzed. I have no qualms about the methods used for such analyses, but I missed some comments (perhaps issued as cautionary notes) about the following:

(1) Surely, rivers, roads, and flyways may act as both barriers and conduits of invading species, depending on their respective biology (e.g., fishes, mammals, birds). I miss comments on this. (2) Countries' borders usually are set along geographic barriers such as seas, lakes, rivers, mountain ranges, high plateaus, icefields; secondarily along biogeographic or ecological barriers such as deserts, forests, steppes. I also miss comments on this. (3) Invasions are often simple range expansions of alien species distributions but more and more often are long-distance translocations facilitated by commerce (trade, travel) or war (another sort of invasion). No mention is made of this distinction. (4) The references (that I imagine served as database sources), are mainly for research published in English, thus providing an unbalanced coverage favoring data inputs from North America and Europe, and to a smaller extent from Asia or Africa. Thus, the lesser importance implicitly attributed to cross-border invasions in South America (where I work and where research is often communicated in Spanish, the second most spoken native language --after Mandarin Chinese-- in the world) may be more an artifact than a fact. I do not expect the authors to be polyglots; I just wish to call attention to this epistemological bias, which is not commented in their paper either. (5) Other epiphenomenal aspects may apply to other continents or regions (I follow the seven-continent model): The Australian/Papuan continent may appear not much invaded because it contains only a handful of countries in comparison to Africa, Asia or Europe. And What about a continent with no countries: Antarctica?

In sum, this is a valuable contribution to a neglected aspect of worldwide interest, not only in terms of scientific research but in the formulation of national public policies and international treaties. A consideration of epistemological aspects such as those proposed above may further this paper's deliverance and stimulate more research on the topic addressed.

Reviewer #2 (Remarks to the Author):

Manuscript by Qing Zhang et al. entitled "One-third of the world's cross-borders are facing high risk of biological invasions" examines the risk of invasion between neighbouring countries taking into account the values assigned to different variables known to influence the introduction (river network density, traffic network density and bilateral trade) and establishment (land use change and richness of established alien vertebrate) of alien species. The risk of introduction and establishment are assessed separately by ranking these variables into high-low values and then selecting the highest ranked variable as the ultimate risk value. Areas presenting the highest risk of introduction and establishment are considered to present the highest level of invasion threat. The manuscript is well written and structured, although some elements should be described in more detail (see specific comment) to facilitate comprehension. I found the overall idea of the manuscript interesting and potentially suitable for future works assessing the risk of invasion between countries, but there are few aspects that limit its applicability as already stated by the authors in the Discussion (L218-231). It would have been appropriate to try to address these limitations in the analysis instead of recognizing them in the manuscript.

MAJOR COMMENTS

-I agree with the authors (as stated in L219-220) that the reliance solely on data from neighbouring areas is a major limitation of the study. In my opinion, the authors could have enhanced the study by considering all bilateral interactions, not just those between neighbouring countries. By focusing solely on neighbouring areas, an essential source of alien propagules is missing. The results present the risk of invasion along country borders, but the data used to evaluate the risk is at the country level and it can be influenced by other factors (such as bilateral trade with non-neighbouring areas).

-I also agree with the authors (as indicated in L222-226) that the use of vertebrates as surrogates to assess propagule pressure of all alien species is critical, because taxonomic data is not uniformly sampled across taxonomic groups. For instance, there is a significant lack of data on invertebrates, which are frequently transported through international trade (not only trade between neighbouring areas). In this regard, I would recommend authors streamline the title and objectives of the study to specifically focus on the risk of vertebrate invasions, rather than encompassing all alien species.

-As summarized in the Results section (L85-94), the risk of invasion along country borders is determined by the spatial overlap of grids associated with various variables known to influence the introduction and establishment of alien species. However, there is no statistical analysis conducted to assess the significance of these variables in explaining the risk of invasion. The importance attributed to these selected variables is primarily based on previous publications, which somewhat restricts the study's novelty.

SPECIFIC COMMENTS

L33: "Land borders" or "country borders"? Land does not have borders, countries do.

L34-37: Simplify this sentence to strengthen the idea that trade and transportation are responsible of biological invasions. Climate change can both have positive and negative effects on alien species, so I would remove all references to climate change.

L37: "making land borders especially vulnerable to IAS" already said in L33. Repetitive.

L38: Revise sentence, cause the meaning of "short distances" is not clear.

L38-39: Neighbouring areas are also susceptible to present similar environmental niches, which might also facilitate the establishment of species present in the other country.

L45: Conflicts and contrasting national policies can play both a positive and negative role in terms of facilitating alien invasions. Conflicts can promote the transportation of alien species through military material, and contrasting national policies can difficult to coordinate IAS management strategies between countries.

L48 & L49 & L71, among others: You are not assessing invasion risk in transboundary areas, but in transboundary neighbouring areas. I think that it would be appropriate to make it clear from the beginning to avoid confusions.

L65-67: This sentence is disconnected from the previous and posterior ones. How is habitat disturbance hypothesis linked to transboundary areas? Sometimes, borders are just fictional lines without clear physical or environmental limits.

L71: Make clear that you are only assessing this risk in terrestrial areas, not in marine ones.

L71 & L91: "spatial overlap analyses" is a broad concept, please be more specific to facilitate comprehension. How were these data combined?

L74-76: Would it be possible to use the full names or some keyword to refer to the variables instead of using the abbreviations?

L76: What do you mean with "across landscape, biotic, and sociopolitical levels"?

L80: I would not say "facilitate", but something like "indicate regions where transboundary collaborations should be a priority".

L85-86: Please, make clear that readers understand what are "pairs of bilateral regions" and the link between these pairs and the grids.

L87-89: The idea of "defined the invasion hotspot in transboundary areas as those top 20% grids with the highest level of risk posed by any one introduction or establishment vector" is ambiguous without reading the Methods section. Is there one single "invasion hotspot" or can these top grids be located in different areas? What constitutes a high level of risk? How is the risk measured? It is

important to understand how this information has been created to validate its utility.

L95-106: How are these results influenced by the length of the borders? Can longer borders have a higher probability of containing a grid with a high level of risk?

L112: Indicate the significance value.

L131-131: How can it be influenced by data biases? Species data has not been sampled systematically across the globe, so it might influence the results to some extent.

L150: Please, provide some information about what you understand by "cross-border cooperation capacity" and "physical barriers" for readers to have some idea about their meaning without having to read the Methods.

L163: What do you mean by "administration-level invasion risk analysis"? Administrations can be at different levels (countries, provinces, etc.). Be more specific please.

L164-166: Please, revise this sentence because there is some circularity in the reasoning.

L218-231: These are limitations, wouldn't they be more appropriate to address them early in the Discussion than here in the Conclusions? In fact, my main concern with this paper is why the authors did not try to address these limitations in the analysis.

L226-228: This is relevant for all countries, not only less developed ones. Invasive alien species have an impact across the globe, so all countries should take care of their spread.

L265-267: It think that it would be good to discuss in more detail the relationship between bilateral trade and local human populations in the Discussion, because it does not seem straightforward to me. The risk of IAS introduction depends on the final destinations of traded goods, but also on the type of commodity being traded and the mode of transportation. Additionally, trade volumes between two countries might not necessarily be transported through cross-border limits but through air transportation.

L267 & L270: What is an "introduction epicenter method" or "trade introduction epicenter"? This concept/method is not clear to me.

L286-287: Did you consider the possibility to include an index related to climatic similarity between the two countries?

L293-299: You are assuming that the distribution of vertebrate alien species is indicative of all alien species, but it would be good to provide some evidences or at least justify potential limitations in more detail.

L318-320: How did you do this analysis?

L327-337: I think that it would be appropriate to have more information on cooperation abilities and physical barriers, particularly the latest. It is hard for me to assess its suitability without knowing how were these indices created. It requires the reader to dig into the original publications.

Reviewer #3 (Remarks to the Author):

The paper used global datasets of invasive species distributions, as well as drivers of alien species introduction and establishment, to identify the hotspots and determinants of invasion risk of cross-border areas. This study is solid and of great importance to biodiversity conservation and sustainable development of border areas. I appreciate the authors' efforts on this interesting topic, which as far as I know, is the first assessment. After reading this manuscript several times, I found the paper is well-written despite that there are several unclear tiny parts in the analysis and the text as I mentioned below, and these problems are easy to address. Please address them before considering publication in Nature Communications.

1. The invasion risk is evaluated based on databases of global distribution maps of invasive species, but the research efforts are biased in developed and well-studied regions such as North America, and Europe, and China, this may influence the observed results: invasion risk in Africa and South America may be underestimated. Although this is a pervasive problem for global studies, the authors need to at least mention this limitation in the discussion.

2. Maybe I missed something, I am not clear of the calculation of invasion risk. In the methods (Line 309-324), the authors mainly talked about the determinants that drive invasion risk. This paragraph needs to be clarified and improved.

3. Physical barrier analysis (Fig. 5). The physical barrier data is mostly from EurAsia (Plos Biology, 2016), I suggest the authors limit the analysis in this area, not at the global scale. In addition, the authors need to only focus on species that will be affected by barriers, all fish species, most birds and amphibians, and many reptiles and small mammal species are unlikely to be influenced by terrestrial barriers.

4. At last, whether the overall invasion risk is influenced by certain taxonomic groups? It is better to show how many bird, mammal, amphibian and reptile species are included in the analysis, and the major groups that contribute the most to the observed patterns.

I have some minor comments below:

Line 16, say this study is mostly based on datasets of global alien species introduction and establishment, not their drivers...

Line 19, most important drivers of?

Line 20-21, need to rewrite based on new analysis.

Line 35, border infrastructure construction, as far as I know, have relatively limited impacts on climate change and greenhouse gas emissions (which are not related to the main topic of this paper either).

Line 51 and 254, please use consistent terms, better replace "human activities" with "human movement", and please check the text thoroughly.

Line 76, please remove "across landscapes, biotic and sociopolitical levels", it is duplicate with former words.

Line 171-176, note country size is also an important factor, species are easy to move neighboring countries if the country is very small with similar biotic and socioeconomic environments, and this phenomenon is common in Europe.

Fig. 1 The quality of this figure needs to be improved.

Fig. 3, it seems the pattern was mostly driven by fish, birds and amphibians, could you please add something in the discussion on which taxa drove the observed pattern (which taxa is the most abundant group, see major comment 4).

**Point-by-Point Response to Reviewers' Comments (MS number: NCOMMS-23-23910A)**

**Reviewer 1**

**Review Comment 1)**

**General comments** Cross-border invasions have been a neglected aspect of such phenomena
because each country takes care of its own problems as if it were completely isolated from neighbors
or from globalized commerce. This is reflected in the lack of coordinated international efforts to
monitor or control alien species that expand their ranges across borders. This paper thus addresses a
novel, synthetic, timely, and welcome addition to the transboundary invasion problem. I am impressed
at the worldwide and multi-species information amassed and analyzed. I have no qualms about the
methods used for such analyses, but I missed some comments (perhaps issued as cautionary notes)
about the following:

**Our Response:** Thank you very much for your positive comments on the novelty and importance of
our study. We have provided detailed responses to each of your comments below.

**Review Comment 2)**

(1) Surely, rivers, roads, and flyways may act as both barriers and conduits of invading species,
depending on their respective biology (e.g., fishes, mammals, birds). I miss comments on this.

**Our Response:** The reviewer makes an excellent point here, and we are grateful for this input on our
manuscript. We completely agree with your opinion that the role of different invasion pathways may
be dependent on the exact taxa. For instance, river network may be particularly important for taxa
such as fish, amphibian and reptile with life stages in the water. So, in our revised manuscript, we
have conducted supplementary analyses on the introduction risk of alien species using the exact
important pathways for certain taxa and addressed this point in the Discussion section (Line 240-243).
We have also clarified this issue in our revised Methods section (Line 344-347). Importantly, our new
analyses obtained similar findings on the nearly one third of crossing borders of neighboring countries
face high risk of alien vertebrate invasions.

**Review Comment 3)**

(2) Countries' borders usually are set along geographic barriers such as seas, lakes, rivers, mountain
ranges, high tableaus, icefields; secondarily along biogeographic or ecological barriers such as
deserts, forests, steppes. I also miss comments on this.

**Our Response:** Thank you very much for this constructive comment on our manuscript. We agree
with your suggestion that geographical barriers could indeed limit the spread of alien species
especially for those taxa with low dispersal abilities. Therefore, in our revised manuscript, we have
conducted a supplementary analysis by incorporating a mountainous topographic heterogeneity index
(MTH) by averaging the maximum range in elevation of all 30 arc-second grid cells within each 0.5°
grids using data from WorldClim followed previous studies (e.g., Jetz et al. 2009, Proceedings B, Liu
et al. 2014 Ecology Letters). We focused on these analyses for most taxa except the birds with
relatively high natural dispersal abilities to overcome the geographic barriers. We have also
addressed this issue in both the revised Discussion (Line 243-246) and Methods section (Line 347-
352).

**Review Comment 4)**

(3) Invasions are often simple range expansions of alien species distributions but more and more
often are long-distance translocations facilitated by commerce (trade, travel) or war (another sort of
invasion). No mention is made of this distinction.

**Our Response:** This is a very important issue and we are sorry that we did not clarify this point in our
original text. However, just as what the editor suggested, our present work focuses on the invasion
risk in transboundary areas of neighboring countries. So, evaluation of all long-distance paired
countries with bilateral trade and species exchanges is beyond the scope of this study. Although we
have addressed this point in our original manuscript (original Line number: 218-222), we further
strengthened this part in our revised Discussion section (Line 260-265) and cited the recent
publication on the role of war in facilitating alien species invasions within and between continents
(Santini et al. 2023 *Frontiers in Ecology and the Environment*). Please also see our response to the
27# comment of Reviewer 2.

**Review Comment 5)**

(4) The references (that I imagine served as database sources), are mainly for research published in
English, thus providing an unbalanced coverage favoring data inputs from North America and Europe,
and to a smaller extent from Asia or Africa. Thus, the lesser importance implicitly attributed to cross-
border invasions in South America (where I work and where research is often communicated in
Spanish, the second most spoken native language --after Mandarin Chinese-- in the world) may be
more an artifact than a fact. I do not expect the authors to be polyglots; I just wish to call attention to
this epistemological bias, which is not commented in their paper either.

**Our Response:** The reviewer makes an excellent point here, and we are grateful for the reviewer's
suggestion on the use of different languages to collect species distributions. In our revised
manuscript, we have re-conducted the literature search using a total of 12 major languages in
literatures including English, French, Danish, Estonian, Finnish, German, Norwegian, Portuguese,
Russian, Spanish, Swedish, and the Mandarin Chinese (please see more details in the supporting
Supplementary Table 2) to collect distribution data of established alien vertebrate across the global
border areas especially for amphibians, reptiles and fish. In fact, some public databases we used
such as the Global Avian Invasions Atlas (GAVIA) and the global Distribution of Alien Mammals
database (DAMA) for established alien bird and mammal species have indeed incorporated the multi-
lingual literatures, which we have also clarified in our Methods section (Line 365-379). In addition, we
have addressed this issue by citing related references (e.g., Zenni et al. 2023 *Journal of Applied
Ecology*) in the revised Methods section (Line 369). Furthermore, we also applied a sampling-effort-
corrected approach (Dawson et al., 2017; Meyer et al., 2015) to test the potential influence of
sampling bias on our results, which also obtained similar results with the main results. All these
analyses demonstrated that our results were robust to potential data bias. Please also see our
response to 21# comment of Reviewer 2 and 2# comment of Reviewer 3.

**Review Comment 6)**

(5) Other epiphenomenal aspects may apply to other continents or regions (I follow the seven-
continent model): The Australian/Papuan continent may appear not much invaded because it contains
only a handful of countries in comparison to Africa, Asia or Europe. And What about a continent with
no countries: Antarctica?

**Our Response:** We are grateful to this constructive comment on the potential effect of the number of
countries on the invasion risks. First, we are sorry for the unclear description that we indeed did not
include Antarctica with no countries, and Australia or New Zealand without terrestrial neighboring
borders in our analyses, which we have clarified in our revised Methods section (Line 298-300).
Second, at the continental level, we also conducted a supplementary Spearman correlation analysis,
and did not detect significantly positive relationship of the length of the border lines and number of
countries in each continent with the number of overall invasion hotspots, which indicated that our
results may not be an artefact only reflecting the different number of countries among continents,
which we have incorporated in the revised Discussion section (Line 206-212).

**Review Comment 7)**

In sum, this is a valuable contribution to a neglected aspect of worldwide interest, not only in terms of
scientific research but in the formulation of national public policies and international treaties. A
consideration of epistemological aspects such as those proposed above may further this paper's
deliverance and stimulate more research on the topic addressed.

**Our Response:** Thank you again for your positive comment on the contribution of our study on
scientific research and management policies. We have tried to conduct additional data collection,
supplementary analyses and strengthen the whole manuscript in response to each of your comments
above in our revised manuscript. We hope that these revisions can satisfy your expectations.

**Reviewer 2**

**General comments** Manuscript by Qing Zhang et al. entitled “One-third of the world’s cross-borders
are facing high risk of biological invasions” examines the risk of invasion between neighbouring
countries taking into account the values assigned to different variables known to influence the
introduction (river network density, traffic network density and bilateral trade) and establishment (land
use change and richness of established alien vertebrate) of alien species. The risk of introduction and
establishment are assessed separately by ranking these variables into high-low values and then
selecting the highest ranked variable as the ultimate risk value. Areas presenting the highest risk of
introduction and establishment are considered to present the highest level of invasion threat. The
manuscript is well written and structured, although some elements should be described in more detail
(see specific comment) to facilitate comprehension. I found the overall idea of the manuscript
interesting and potentially suitable for future works assessing the risk of invasion between countries,
but there are few aspects that limit its applicability as already stated by the authors in the Discussion
(L218-231). It would have been appropriate to try to address these limitations in the analysis instead
of recognizing them in the manuscript.

**Our Response:** Thank you very much for your interests with the topic and finding it potentially useful
for assessing future risk of invasion between countries. We apologize for the unclear descriptions on
the details of our manuscript, which we have revised following each of your suggestions below.

**MAJOR COMMENTS:**

**Review Comment 1)** I agree with the authors (as stated in L219-220) that the reliance solely on data
from neighbouring areas is a major limitation of the study. In my opinion, the authors could have
enhanced the study by considering all bilateral interactions, not just those between neighbouring
countries. By focusing solely on neighbouring areas, an essential source of alien propagules is
missing. The results present the risk of invasion along country borders, but the data used to evaluate
the risk is at the country level and it can be influenced by other factors (such as bilateral trade with
non-neighbouring areas).

**Our Response:** The reviewer makes an excellent point here. We are sorry for the unclear description
in our original manuscript on this important issue. Our present work focuses on the invasion risk at
border areas of neighboring countries, which is in theoretically under high risk of biological invasions
due to the short distance, similar climate and frequent trade and economic exchanges but is still a
neglected aspect in invasion literature. So, evaluation of all paired countries is important but indeed
beyond the scope of this study. We therefore have followed the editor’s suggestion by keeping the
neighboring areas in our revised manuscript and have further strengthened the discussion by
stressing this issue in the revised Discussion section (Line 260-265).

**Review Comment 2)** -I also agree with the authors (as indicated in L222-226) that the use of
vertebrates as surrogates to assess propagule pressure of all alien species is critical, because
taxonomic data is not uniformly sampled across taxonomic groups. For instance, there is a significant
lack of data on invertebrates, which are frequently transported through international trade (not only
trade between neighbouring areas). In this regard, I would recommend authors streamline the title

and objectives of the study to specifically focus on the risk of vertebrate invasions, rather than
encompassing all alien species.

**Our Response:** We are sorry for this unclear point. Following the reviewer and the editor's
suggestion, we have changed the title to focus on the risk of alien vertebrate invasions and made
updates throughout the whole text. We have also further strengthened the Discussion section as one
potential limitation on the taxa we used in the present study (Line 270-273).

**Review Comment 3)** As summarized in the Results section (L85-94), the risk of invasion along
country borders is determined by the spatial overlap of grids associated with various variables known
to influence the introduction and establishment of alien species. However, there is no statistical
analysis conducted to assess the significance of these variables in explaining the risk of invasion. The
importance attributed to these selected variables is primarily based on previous publications, which
somewhat restricts the study's novelty.

**Our Response:** Thank you very much for this excellent suggestion. In our revised manuscript, we
have tried to conduct statistical analyses from two aspects. First, we compared the proportion of high-
risk grids for each of predictor variables along global transboundary borders using Kruskal-Wallis test.
Second, we applied a null model approach to test whether the spatial overlap between introduction
and establishment risks were more concentrated in certain regions than that would be expected by
chance based on Mann-Whitney U test. As you will see, our new analyses find that BTV, LUCF and
REA are still the most important factor predicting the high overall invasion risk of neighboring areas
(Kruskal-Wallis test, $H=110.58$, $P < 0.01$, Supplementary Fig. 3). By generating 1000 random
overlaps of high introduction and high establishment grids, null model test shows that there is a
significant overlap among important predictor variables than expected by chance based on null
distributions (Mann-Whitney U test: $Z = -1.73$, $P < 0.05$). We have added these new analyses in both
revised Methods (Line 405-408, 410-413) and Results sections (Line 132-138).

**SPECIFIC COMMENTS**

**Review Comment 4)** L33: "Land borders" or "country borders"? Land does not have borders,
countries do.

**Our Response:** Sorry for the misunderstanding, we have changed land borders to country borders
throughout the whole text.

**Review Comment 5)** L34-37: Simplify this sentence to strengthen the idea that trade and
transportation are responsible of biological invasions. Climate change can both have positive and
negative effects on alien species, so I would remove all references to climate change.

**Our Response:** We have rephrased the sentences to make it more succinct and clearer (Line 36-39).
Following the reviewer's suggestion, we have removed all sentences and references related with
climate change in our revised text.

**Review Comment 6)** L37: "making land borders especially vulnerable to IAS" already said in L33.
Repetitive.

**Our Response:** Thank you for this suggestion. We have removed this description here and made it
more clear to readers (Line 39).

**Review Comment 7)** L38: Revise sentence, cause the meaning of “short distances” is not clear.

**Our Response:** Sorry for the unclear description. We have tried to clarify this sentence following the
reviewer’s suggestion (Line 39-40).

**Review Comment 8)** L38-39: Neighbouring areas are also susceptible to present similar
environmental niches, which might also facilitate the establishment of species present in the other
country.

**Our Response:** Thank you for this helpful suggestion. We completely agree with the reviewer that a
similar environmental niche between neighboring countries may facilitate the establishment of non-
native species from one side to another. However, just as what the previous study suggested (e.g.,
Early et al. 2016), although environmental suitability is important to non-native species establishment,
different species may have different environmental niche requirements and thus this variable is
always assessed by species-specific studies. Therefore, as the information on which species has
entered into different countries and their environmental requirements is not available yet at the global
scale, we did not include this variable in our analysis. In our revised text, we have addressed this
issue in the Introduction section (Line 43-45) by stressing the important effect of environmental niche
similarity in driving invasion risks between neighboring countries and cited related supporting
reference, we also discussed this as one potential avenue for future studies when related data is
available worldwide (Line 266-270).

**Review Comment 9)** L45: Conflicts and contrasting national policies can play both a positive and
negative role in terms of facilitating alien invasions. Conflicts can promote the transportation of alien
species through military material, and contrasting national policies can difficult to coordinate IAS
management strategies between countries.

**Our Response:** Thank you for this very detailed suggestion on the complicated effect of conflict and
national policy on invasion risks. We have added these very helpful suggestions in our revised
Introduction section (Line 47-52). Specific to the role of conflict such as war in biological invasions, we
have also cited the recent publication (Santini et al. 2023 *Frontiers in Ecology and the Environment*) in
our revised manuscript (Line 50).

**Review Comment 10)** L48 & L49 & L71, among others: You are not assessing invasion risk in
transboundary areas, but in transboundary neighbouring areas. I think that it would be appropriate to
make it clear from the beginning to avoid confusions.

**Our Response:** We are sorry for the unclear description, which we have tried to make it clearly
throughout the whole manuscript (e. g. Line 53-56).

**Review Comment 11)** L65-67: This sentence is disconnected from the previous and posterior ones.
How is habitat disturbance hypothesis linked to transboundary areas? Sometimes, borders are just
fictional lines without clear physical or environmental limits.

**Our Response:** We apologize for this unclear description that may have confused you. As you will
see, based on the literature review, we found that crossing border areas are also facing increasing
pressures from anthropogenic disturbances (Liu et al. 2020 Trends Ecol Evol 35: 679-690.). In our
revised text, we have tried to clarify this point to not confuse readers (Line 72-73).

**Review Comment 12)** L71: Make clear that you are only assessing this risk in terrestrial areas, not in
marine ones.

**Our Response:** We have made it clearly in our revised text that our present study only assessed the
risk in terrestrial areas but not in marine ones here (Line 85).

**Review Comment 13)** L71 & L91: “spatial overlap analyses” is a broad concept, please be more
specific to facilitate comprehension. How were these data combined?

**Our Response:** We are sorry for this unclear description on the spatial overlap analysis. We have
provided more details on the process we conducted the overlap analysis among different predictors,
and between introduction risk and establishment risk in our revised manuscript both in the
Introduction (Line 85-87) and the Methods sections (Line 397-401).

**Review Comment 14)** L74-76: Would it be possible to use the full names or some keyword to refer to
the variables instead of using the abbreviations?

**Our Response:** Sorry for the misunderstanding. We have followed the reviewer’s suggestion by
providing the full names instead of the abbreviations to make it more clearly to readers (Line 89-91).

**Review Comment 15)** L76: What do you mean with “across landscape, biotic, and sociopolitical
levels”?

**Our Response:** Sorry for this misunderstanding. We have removed this unclear and redundant
sentence in our revised manuscript to not confuse readers. Please also see our response to the 11#
comment of Reviewer 3.

**Review Comment 16)** L80: I would not say “facilitate”, but something like “indicate regions where
transboundary collaborations should be a priority”.

**Our Response:** We have modified this sentence as the reviewer suggested (Line 94-95).

**Review Comment 17)** L85-86: Please, make clear that readers understand what are “pairs of
bilateral regions” and the link between these pairs and the grids.

**Our Response:** Sorry for the misunderstanding. We have provided the definition of pairs of bilateral
regions with more methodological details here (Line 100-101).

**Review Comment 18)** L87-89: The idea of “defined the invasion hotspot in transboundary areas as
those top 20% grids with the highest level of risk posed by any one introduction or establishment
vector” is ambiguous without reading the Methods section. Is there one single “invasion hotspot” or
can these top grids be located in different areas? What constitutes a high level of risk? How is the risk
measured? It is important to understand how this information has been created to validate its utility.

**Our Response:** We are sorry for not providing a clear description on how we determined the invasion
hotspot in our original Results section. In our revised text, we have provided the detailed process on
how we calculate the invasion hotspots in the Results section (Line 102-106).

**Review Comment 19)** L95-106: How are these results influenced by the length of the borders? Can
longer borders have a higher probability of containing a grid with a high level of risk?

**Our Response:** Thank you for this very constructive comment. In order to test whether a longer
border line will lead to a higher number of grids with high invasion risk, we have conducted a
Spearman correlation analysis by comparing the relationship between the length of the borders and
the number of overall invasion hotspots (grids with VH level overall risk) across continents. In
addition, we have also tested whether there might be positive relationship of the number of high-risk
grids with area and number of neighboring countries among continents. Our analysis did not detect a
significant relationship between the length of the borders and the number of overall invasion hotspots
(Spearman correlation coefficient $r=0.5$, $P>0.05$), indicating that our findings should not be an artefact
resulting from longer borders, more and larger countries among continents. We have addressed this
important issue in our revised Discussion section of the manuscript (Line 206-212). Please also see
our response to a similar comment by Reviewer 1 (6# comment) and Reviewer 3 (12# comment).

**Review Comment 20)** L112: Indicate the significance value.

**Our Response:** We have re-conducted the statistical analyses using a null-model approach and
added the significance value in our revised Results section (Line 136-138). Please also see our
response to the reviewer's 3# comment above.

**Review Comment 21)** L131-131: How can it be influenced by data biases? Species data has not
been sampled systematically across the globe, so it might influence the results to some extent.

**Our Response:** Thank you for this very important suggestion on potential sampling bias. We
completely agree with you that there are indeed variations in sampling bias among taxa and
geographical ranges. For example, there would be more sampling effort in countries with stronger
research abilities such as those North American and European countries. Countries in south America,
Africa and Asia may be less sampled than those continents. In our revised manuscript, we have
followed the suggestion of Reviewer 1 by re-searching the species data using a total of 12 languages
especially for herpetofauna and fish. For birds and mammals, the dataset we used have incorporated
information with different languages and have been widely used in the literature. Furthermore, we
applied a sampling-effort-corrected approach (Dawson et al., 2017; Meyer et al., 2015) to test the
potential influence of sampling bias on our results, which also obtained similar results with the main
results. We have added related sentences on this issue in the revised Results (Line 158-163) and
Methods sections (Line 421-426). Finally, we have strengthened this potential limitation in the
Discussion section (Line 237-240). Please also see our responses to 5# comment of Reviewer 1 and
2# comment of Reviewer 3.

**Review Comment 22)** L150: Please, provide some information about what you understand by "cross-

border cooperation capacity” and “physical barriers” for readers to have some idea about their
meaning without having to read the Methods.

**Our Response:** We are sorry for the unclear descriptions on “cross-border cooperation capacity” and
“physical barriers”, which we have clarified in the revised Results section before the Methods (Line
180-183).

**Review Comment 23)** L163: What do you mean by “administration-level invasion risk analysis”?
Administrations can be at different levels (countries, provinces, etc.). Be more specific please.

**Our Response:** We are sorry for this unclear information. What we mean here is regarding previous
invasion risk analysis not focusing on the transboundary neighboring areas but usually worked on the
whole country level. We have tried to modify this sentence to not confuse readers (Line 195-196).

**Review Comment 24)** L164-166: Please, revise this sentence because there is some circularity in
the reasoning.

**Our Response:** Sorry for the misunderstanding. We have modified this sentence to avoid the
circularity problem (Line 194-199).

**Review Comment 25)** L218-231: These are limitations, wouldn't they be more appropriate to address
them early in the Discussion than here in the Conclusions? In fact, my main concern with this paper is
why the authors did not try to address these limitations in the analysis.

**Our Response:** Thank you again for this helpful suggestion. In response to your concerns on those
potential limitations, we have re-written this part and appear earlier before the conclusion in the
Discussion section (Line 260-278). Importantly, we have also tried to address some limitations such
as sampling bias in our new round of data collection and analyses, and discussed the other aspects
of limitations as future potential study directions.

**Review Comment 26)** L226-228: This is relevant for all countries, not only less developed ones.
Invasive alien species have an impact across the globe, so all countries should take care of their
spread.

**Our Response:** Thank you for this reasonable suggestion. We have rephrased this sentence by
stressing the anthropogenic spread of alien species all over the world (Line 273-275).

**Review Comment 27)** L265-267: It think that it would be good to discuss in more detail the
relationship between bilateral trade and local human populations in the Discussion, because it does
not seem straightforward to me. The risk of IAS introduction depends on the final destinations of
traded goods, but also on the type of commodity being traded and the mode of transportation.
Additionally, trade volumes between two countries might not necessarily be transported through
cross-border limits but through air transportation.

**Our Response:** Thank you for this constructive suggestion. In our revised manuscript, we have
collected the number of airline routes for airports in each grid of the transboundary neighboring areas
as one new variable and conducted supplementary analysis predicting the introduction risk of alien

species through air transportation. We have also added new descriptions on this issue in the Methods
section (Line 336-339).

**Review Comment 27)** L267 & L270: What is an “introduction epicenter method” or “trade introduction
epicenter”? This concept/method is not clear to me.

**Our Response:** We are sorry for this unclear description on the methods we used to quantify the
invasion epicenter along transboundary neighboring areas. We have re-written this part using more
precise and concise descriptions in our revised Methods section (Line 320-327).

**Review Comment 28)** L286-287: Did you consider the possibility to include an index related to
climatic similarity between the two countries?

**Our Response:** Thank you for this suggestion. Please also see our response to the reviewer’s 8#
comment above. Considering that different species may have different climatic niches and thus this
variable is always assessed by species-specific studies as suggested by the previous study (Early et
al. 2016), we did not include this variable in our analysis. In our revised text, we have addressed this
issue in both the Introduction (Line 77-80) and Discussion section (Line 266-270).

**Review Comment 29)** L293-299: You are assuming that the distribution of vertebrate alien species is
indicative of all alien species, but it would be good to provide some evidences or at least justify
potential limitations in more detail.

**Our Response:** Thank you a lot for this important issue. Please also see our response to the
reviewer’s 2# comment above. In addition to acknowledge this potential taxonomic limitation in the
revised Discussion section (Line 270-273), we have also changed our title to focus on the risk of
introduction and establishment of alien vertebrate according to the reviewer and the editor’s
suggestion.

**Review Comment 30)** L318-320: How did you do this analysis?

**Our Response:** We apologize for the unclear descriptions on the methods we used to quantify the
relative importance of different introduction and establishment factors in predicting the invasion risks
along transboundary neighboring areas. In our original manuscript, we only compared the number of
different grids with highest risks for each predictor. In our revised manuscript, following your
suggestion, we have strengthened the statistical analyses by using the Kruskal-Wallis test to explore
the relative importance of different introduction and establishment factors in predicting overall invasion
hotspots along various borders (Line 405-408). We also applied a null-model approach to test
whether the observed overlap among different predictor variables is more concentrated on some
regions than that with the expected by chance based on null distributions (Line 410-413). Please also
see our response to your 3# comment above.

**Review Comment 31)** L327-337: I think that it would be appropriate to have more information on
cooperation abilities and physical barriers, particularly the latest. It is hard for me to assess its
suitability without knowing how these indices were created. It requires the reader to dig into the
original publications.

**Our Response:** We are sorry that we did not provide the detailed information on the definition and
quantification of cooperation abilities and physical barriers. In our revised manuscript, we have
provided more details on how we used the two metrics in our revised Methods section (Line 434-439).

**Reviewer 3**

**General comment 1)** The paper used global datasets of invasive species distributions, as well as
drivers of alien species introduction and establishment, to identify the hotspots and determinants of
invasion risk of cross-border areas. This study is solid and of great importance to biodiversity
conservation and sustainable development of border areas. I appreciate the authors' efforts on this
interesting topic, which as far as I know, is the first assessment. After reading this manuscript several
408 times, I found the paper is well-written despite that there are several unclear tiny parts in the analysis
and the text as I mentioned below, and these problems are easy to address. Please address them
before considering publication in Nature Communications.

**Our Response:** Thank you very much for your positive and constructive comments on our
manuscript. We have addressed all those unclear parts in the analysis and the text below following
your suggestions.

**Review Comment 2)** 1. The invasion risk is evaluated based on databases of global distribution
maps of invasive species, but the research efforts are biased in developed and well-studied regions
such as North America, and Europe, and China, this may influence the observed results: invasion risk
in Africa and South America may be underestimated. Although this is a pervasive problem for global
studies, the authors need to at least mention this limitation in the discussion.

**Our Response:** We appreciate this excellent suggestion very much. We completely agree that there
is indeed sampling bias across both taxonomic groups and geographic regions. Consequently, in our
revised manuscript, we have followed the 5# suggestion of Reviewer 1 by re-conducting data
collection on the global distributions of established alien species using more languages especially for
fish, amphibians and reptiles because these taxa have relatively lower sampling effort than invasive
birds and mammals that have published in the journals Scientific Data (alien avian database) and
Ecology (alien mammal dataset), and have incorporated multilingual literatures. In addition, we have
also added one new analysis by applying a sampling-effort-corrected approach (Dawson et al., 2017;
Meyer et al., 2015) to test the potential influence of sampling bias on our results, which also obtained
similar results with the main results. We have also followed the reviewer's suggestion by addressing
this issue in the revised Discussion section (Line 237-240). Please also see our response to 5#
comment of Reviewer 1 and 21# comment of Reviewer 2.

**Specific comments:**

**Review Comment 3)** 2. Maybe I missed something, I am not clear of the calculation of invasion risk.
In the methods (Line 309-324), the authors mainly talked about the determinants that drive invasion
risk. This paragraph needs to be clarified and improved.

**Our Response:** We are sorry for the unclear description on the calculation of invasion risk. We have
tried to strengthen this part by providing more details in the calculation process of invasion risk in both
the revised text (Line 397-401) and the revised Supplementary Fig. 1 (Line 694).

**Review Comment 4)** 3. Physical barrier analysis (Fig. 5). The physical barrier data is mostly from
EurAsia (Plos Biology, 2016), I suggest the authors limit the analysis in this area, not at the global
scale. In addition, the authors need to only focus on species that will be affected by barriers, all fish

species, most birds and amphibians, and many reptiles and small mammal species are unlikely to be
influenced by terrestrial barriers.

**Our Response:** Thank you for this very helpful suggestion. We have followed your suggestion by
conducting two supplementary analyses specific to Eurasia transboundary neighboring area and non-
native reptiles and mammals that may be more easily affected by physical barriers (Line 440-444).
Therefore, after the global scale analysis in the main text, we also conducted supplementary analyses
specific to the Eurasia areas (Supplementary Fig. 14) and the alien reptile and mammal species
(Supplementary Fig. 15), respectively.

**Review Comment 5)** 4. At last, whether the overall invasion risk is influenced by certain taxonomic
groups? It is better to show how many bird, mammal, amphibian and reptile species are included in
the analysis, and the major groups that contribute the most to the observed patterns.

**Our Response:** Thank you for pointing this out. In our revised manuscript, we have provided more
detailed information on the exact number of bird, mammal, amphibian and reptile species included in
the analysis in the revised Methods section (Line 365), and clarified the exact taxonomic groups that
contribute most to the observed invasion risks in the revised Results section (Line 154-156).

**Review Comment 6)** Line 16, say this study is mostly based on datasets of global alien species
introduction and establishment, not their drivers...

**Our Response:** We are thankful for this suggestion and have made this point clearly in the revised
Abstract section (Line 16).

**Review Comment 7)** Line 19, most important drivers of?

**Our Response:** Sorry for the unclear point. We have clarified this point by describing the important
predictor variables of the introduction and establishment risks of alien vertebrate species (Line 19-20).

**Review Comment 8)** Line 20-21, need to rewrite based on new analysis.

**Our Response:** Thanks. We have conducted supplementary analysis specific to the Eurasia area
shown in Supplementary Fig. 14 according to your 4# comment above (Line 440-444). As you will
see, the low overlap between invasion hotspots and border fences are still low in Eurasia where the
border fences mainly distribute (Line 23).

**Review Comment 9)** Line 35, border infrastructure construction, as far as I know, have relatively
limited impacts on climate change and greenhouse gas emissions (which are not related to the main
topic of this paper either).

**Our Response:** Thank you very much for pointing out this issue. We completely agree and have
removed this sentence in our revised text (Line 38). Please also see our response to the 5# comment
of Reviewer 2.

**Review Comment 10)** Line 51 and 254, please use consistent terms, better replace "human
activities" with "human movement", and please check the text thoroughly.

**Our Response:** Thank you. We have used human movement throughout the whole text following
your suggestion (Line 58 & 309 & 314).

**Review Comment 11)** Line 76, please remove “across landscapes, biotic and sociopolitical levels”, it
is duplicate with former words.

**Our Response:** Sorry for this redundant description and we have removed this sentence in the
Introduction section (Line 91). Please also see our response to the 15# comment of Reviewer 2.

**Review Comment 12)** Line 171-176, note country size is also an important factor, species are easy
to move neighboring countries if the country is very small with similar biotic and socioeconomic
environments, and this phenomenon is common in Europe.

**Our Response:** Thank you for this excellent suggestion on the potential reasons why we observed a
high invasion risk in Europe. To address this potential explanation, we conduct a supplementary
analysis by investigating the relationship between the number of overall invasion hotspots (grids with
VH level overall risk) and the country size at continent level. But we did not find a positive relationship
between these two variables (Spearman correlation coefficient $r = 0.3$, $P > 0.05$), indicating that our
results could not be influenced by the country size. We have also explored the number of high-risk
grids with the border length and the number of countries involved, which also did not detect significant
relationships. We have addressed this helpful comment in the revised Discussion section (Line 206-
212). Please also see our response to 6# comment of Reviewer 1 and 12# comment of Reviewer 2.

**Review Comment 13)** Fig. 1 The quality of this figure needs to be improved.

**Our Response:** Sorry for the unclear figure. In the revised manuscript, we have shown the risk of
alien species introduction and establishment in Figure 1, and moved the overall risk of invasion to a
new Figure 2. This can help us increase the spatial resolution and quality of the figure (Line 650 &
657).

**Review Comment 14)** Fig. 3, it seems the pattern was mostly driven by fish, birds and amphibians,
could you please add something in the discussion on which taxa drove the observed pattern (which
taxa is the most abundant group, see major comment 4).

**Our Response:** Thank you for this very helpful suggestion. We have conducted supplementary
analysis by showing the relative proportion of high-risk grids for the establishment of alien vertebrates
for each taxonomic group, and provided the result in a new Supplementary Fig. 7 and Supplementary
Data 4. We have also addressed this issue in our revised Results section (Line 154-156) following the
reviewer's suggestion.

REVIEWERS' COMMENTS

Reviewer #1 (Remarks to the Author):

This is my second time around this manuscript. I notice that my previous suggestions have been duly addressed and I thank the authors for taken them seriously and having corrected/amended/complemented their analyses and the prose rendition of this new version. I also notice that the authors correctly responded to the criticisms of the other two reviewers. Thus, I feel satisfied with the new product. It is a timely and welcome addition to the arsenal of biological invasions knowledge with reference to transboundary phenomena at a planetary scale.

Reviewer #3 (Remarks to the Author):

I have read the response letter, both to my comments and that to the other reviewers, and I found the authors have done a great job in addressing the concerns. I have no further comments to add.

Point-by-Point Response to Reviewers' Comments (MS number: NCOMMS-23-23910A)

Reviewer 1

Review Comment

This is my second time around this manuscript. I notice that my previous suggestions have been duly addressed and I thank the authors for taken them seriously and having corrected/amended/complemented their analyses and the prose rendition of this new version. I also notice that the authors correctly responded to the criticisms of the other two reviewers. Thus, I feel satisfied with the new product. It is a timely and welcome addition to the arsenal of biological invasions knowledge with reference to transboundary phenomena at a planetary scale.

Our Response: Thank you very much for your positive comments on our revisions, and finding our work timely and useful to prevent biological invasions at transboundary areas.

Reviewer 3

Review Comment

I have read the response letter, both to my comments and that to the other reviewers, and I found the authors have done a great job in addressing the concerns. I have no further comments to add.

Our Response: Thank you very much for your positive comments on our revisions.